# Tilting the Balance: Therapeutic Prospects of CD83 as a Checkpoint Molecule Controlling Resolution of Inflammation

**DOI:** 10.3390/ijms23020732

**Published:** 2022-01-10

**Authors:** Katrin Peckert-Maier, Dmytro Royzman, Pia Langguth, Anita Marosan, Astrid Strack, Atefeh Sadeghi Shermeh, Alexander Steinkasserer, Elisabeth Zinser, Andreas B. Wild

**Affiliations:** Department of Immune Modulation, Universitätsklinikum Erlangen, Friedrich-Alexander Universität—Erlangen-Nürnberg, 91052 Erlangen, Germany; dmytro.royzman@uk-erlangen.de (D.R.); pia.langguth@uk-erlangen.de (P.L.); anita.marosan@uk-erlangen.de (A.M.); astrid.strack@uk-erlangen.de (A.S.); atefeh.sadeghishermeh@uk-erlangen.de (A.S.S.); alexander.steinkasserer@uk-erlangen.de (A.S.); elisabeth.zinser@uk-erlangen.de (E.Z.)

**Keywords:** CD83, resolution of inflammation, IDO/TGF-β-axis, pro-resolving macrophages, adoptive transfer, transplantation

## Abstract

Chronic inflammatory diseases and transplant rejection represent major challenges for modern health care. Thus, identification of immune checkpoints that contribute to resolution of inflammation is key to developing novel therapeutic agents for those conditions. In recent years, the CD83 (cluster of differentiation 83) protein has emerged as an interesting potential candidate for such a “pro-resolution” therapy. This molecule occurs in a membrane-bound and a soluble isoform (mCD83 and sCD83, respectively), both of which are involved in resolution of inflammation. Originally described as a maturation marker on dendritic cells (DCs), mCD83 is also expressed by activated B and T cells as well as regulatory T cells (Tregs) and controls turnover of MHC II molecules in the thymus, and thereby positive selection of CD4^+^ T cells. Additionally, it serves to confine overshooting (auto-)immune responses. Consequently, animals with a conditional deletion of CD83 in DCs or regulatory T cells suffer from impaired resolution of inflammation. Pro-resolving effects of sCD83 became evident in pre-clinical autoimmune and transplantation models, where application of sCD83 reduced disease symptoms and enhanced allograft survival, respectively. Here, we summarize recent advances regarding CD83-mediated resolution of inflammatory responses, its binding partners as well as induced signaling pathways, and emphasize its therapeutic potential for future clinical trials.

## 1. Introduction

Inflammation is a tightly controlled mechanism that ensures an effective response of our organism to potential injuries. Tissue damage, either sterile or after infection, leads to the release of exogenous or endogenous danger signals, which trigger the initial induction phase of inflammation. Tissue resident cells, such as macrophages (Mφ), sense an inflammatory stimulus via pattern-recognition receptors, such as toll-like receptors (TLRs), which causes activation of pro-inflammatory signaling cascades, such as the nuclear factor kappa B (NF-κB) pathway [1]. This activation culminates in secretion of pro-inflammatory cytokines and chemokines, which then orchestrate a sequential recruitment of circulating immune cells to initiate an inflammatory response.

The first cellular response team is formed by neutrophil granulocytes, which invade the inflamed tissue in a swarm-like fashion and potentiate inflammation [2]. Neutrophils also recruit circulating inflammatory monocytes that differentiate into Mφ, which clear potential pathogens as well as cellular debris [1,3]. Either the effector cells of the innate immune system suffice to clear the cause of activation or call on the adaptive immune system for aid. For instance, dendritic cells (DCs), which are the most potent antigen-presenting cells of the immune system, mature upon encounter with inflammogens and migrate to the lymph nodes where they activate naïve T cells, which subsequently differentiate into effector T cells. DCs function as a control center at the interface between innate and adaptive immunity, fine-tuning initiation and eventual confinement of inflammatory processes. Furthermore, specific subsets of these cells promote and maintain immunological tolerance by induction of regulatory T cells (Treg), and thus can restrain inflammation [4].

Once the original harmful stimulus has been successfully cleared, it is of utmost importance to confine the inflammatory reaction to prevent excessive tissue damage. This phase is termed resolution of inflammation and it depends on a switch of specific gene expression profiles towards anti-inflammatory mediators and tissue remodeling [5,6]. One key event in the transition to the resolution phase is reprogramming of Mφ phenotype and function [7]. After the first infiltration wave, neutrophils eventually undergo apoptosis, for instance after phagocytosis of pathogenic threats [8]. During the initial inflammatory response, recruited pro-inflammatory Mφ ingest apoptotic neutrophils via a process called efferocytosis. In contrast to phagocytosis of cellular debris or pathogens, efferocytosis of neutrophils modulates Mφ activation and switches Mφ from an inflammatory to an anti-inflammatory and pro-resolving gene profile [9]. In addition, Tregs that have infiltrated the inflamed tissue can skew the Mφ phenotype to a reparative one [6,7]. Pro-inflammatory Mφ are often termed as classically activated cells (CAM or M1-like), whereas reparative Mφ are referred to as alternatively activated (AAM or M2-like), which is a correlate of stimulation with LPS+IFN-γ or IL4, respectively. However, it has become increasingly clear that the rigid M1/M2 dichotomy is an in vitro phenomenon and not sufficient to describe the entire complexity of Mφ polarization [10]. Mφ rather integrate several environmental cues to adopt the perfect phenotype that is needed to promote a regular course of inflammation—including the resolution phase [7]. The functional switch of Mφ is further supported by a finely concerted sequential expression of pro- and anti-inflammatory cytokines and growth factors. While TNF-α and IFN-γ dominate the initial phase of inflammation, resolution of inflammation is carried by expression of IL-10, IGF-1, and TGF-β [11,12,13].

Since a precise succession of initiation and resolution phase is a prerequisite for eliminating the inflammogen without causing immoderate damage, perturbation of this process can result in chronification of inflammation [6]. For instance, autoimmune disorders, such as rheumatoid arthritis (RA) or multiple sclerosis (MS), show a markedly disturbed resolution of inflammation due to constant triggering of immune responses by the respective autoantigen(s) [14,15]. Similarly, impaired resolution of inflammation fosters graft rejection after transplantation [16]. Treatment of such disorders, where proper shutdown of inflammation is disturbed, has relied on administration of broadly acting immunosuppressive drugs for a long time. However, this kind of treatment often comes with severe side effects, since it not only prevents unwanted but also desired immune responses, and even novel biologicals, such as antibodies against TNF-α, IL-6, or IL-12, bear the risk of an adverse impact on immune reactions [6]. Thus, current research focuses on deciphering molecular checkpoints and signaling pathways, whose modulation would rather directly promote resolution of inflammation than simply suppress the natural process of inflammation [17]. An ideal candidate compound for this purpose would be an endogenous modulator to minimize anti-drug responses, and it should not blunt the initiation of inflammation but rather bolster the resolution phase. Recent research revealed that the CD83 molecule possesses these qualities and is therefore perfectly suited as a putative immunomodulatory agent for future treatment of chronic inflammatory diseases. Therefore, this review article will focus on the role of CD83 as an immunologic checkpoint molecule and will discuss future prospects of its use for “pro-resolution-therapies”.

## 2. CD83: From Maturation Marker to Pro-Resolving Checkpoint Molecule

Since its discovery in 1992 as a surface molecule on activated immune cells such as DCs and B cells, the CD83 molecule has been intensively studied and characterized [18,19,20]: the CD83 protein is highly conserved among distinct species as murine, and human CD83 share 63% amino acid identity [21,22]. Membrane-bound CD83 (mCD83) is extensively glycosylated, which almost doubles its theoretical molecular weight from 23 kDa to 45 kDa, and it consists of three domains: an extracellular Ig-like domain, a transmembrane domain and a cytoplasmic domain [18]. A soluble isoform (sCD83) that consists largely of the extracellular domain is released into the supernatant of activated DCs and B cells [23]. This isoform is generated either by alternative splicing or by proteolytic cleavage of mCD83 [23,24]. The sCD83 molecule is present in low levels in sera of healthy donors, but its abundancy increases in sera of patients with hematological malignancies, e.g., multiple myeloma and acute myeloid leukemia, where it inversely correlates with progression-free survival [25]. In addition, patients with chronic inflammatory diseases, such as multiple sclerosis [26] and rheumatoid arthritis [27] show elevated sCD83 serum levels.

Originally described as a marker for mature DCs (mDCs) [22,28,29,30,31], CD83 expression has been demonstrated on many other activated immune cells, including neutrophils [32], monocytes, macrophages (Mφ) [33], B/T cells [34], NK cells [35], and Tregs [36]. Moreover, also non-immune cells like epithelial cells of the thymus, airways, and intestine express CD83 [37,38,39]. Interestingly, CD83 is also a target of immune escape mechanisms. For instance, infection of DCs with herpes viruses results in proteasomal degradation of CD83, which leads to inhibition of potent antiviral immune responses [40,41,42]. Furthermore, Hodgkin lymphoma cells express CD83 to subvert anti-tumor T cell responses, in part by secretion of sCD83 [43].

In the past few years, various studies have elucidated the biological role of the mCD83 molecule in the context of homeostasis and immune pathologies. Within these studies, the mCD83 protein has emerged as a master regulator for CD4^+^ T cell development [44], and recently mCD83 was characterized as an immunoregulatory molecule, which contributes to maintenance of tolerance [30,36]. Moreover, the sCD83 isoform possesses striking capacities to induce the resolution of inflammation, shown in different pre-clinical models for chronic inflammatory/autoimmune diseases, food allergy and transplantation [39,45,46,47,48,49,50,51]. Within the following sections, we will summarize current knowledge on CD83 elicited signaling events and its pro-resolving function in homeostasis, autoimmune pathologies as well as in transplantation.

### 2.1. Biological Function of CD83 and Its Induced Signaling Events

For many years since its discovery, the signaling capacity of CD83 has remained enigmatic owing to the following problems: (i) the cytoplasmic tail of mCD83 lacks consensus motifs, which allow for binding of adaptor and signaling molecules, and (ii) CD83 has long been received as kind of an “orphan” receptor. While the latter issue was partially solved by identification of binding partners for sCD83 (see below), mCD83 signaling is still the subject of intensive investigations. In this section, we will focus on results demonstrating the importance of mCD83 for inflammation and then discuss still existing gaps in our knowledge regarding its signaling. The subsequent section will then deal with the immunomodulatory signaling capacity of sCD83.

#### 2.1.1. Biological Relevance of mCD83—Beware the Ides of MARCH

Due to its induction of activated immune cells, mCD83 was suggested to exert co-stimulatory functions for a long time. Surprisingly though, data from genetically engineered mice did not corroborate this notion, since APCs from CD83^−/−^ mice have a similar capacity to stimulate antigen-specific T cells or are even more potent in doing so [52,53]. Conditional deletion on DCs further demonstrated a pro-resolving function of CD83 during inflammation (see below) [29,30]. Notwithstanding, studies on complete CD83^−/−^ mice revealed a central signaling activity of CD83, highlighting its importance in the immune system: animals that lack expression of CD83 show a drastic reduction of peripheral CD4^+^ T cells, and deletion of CD83 either systemically or in the respective cell types diminishes surface display of MHC-II and CD86 in DCs and B cells [44,53,54,55]. A similar observation was made in mice that bore a chemically induced mutation, which abolished CD83 surface expression [31]. This study also made the seminal finding that reduced surface display of MHC-II and CD86 due to a lack of CD83 in DCs was rescued upon re-introduction of the TM-domain of CD83 into mutant cells. By mutating different parts of CD83, researchers identified the transmembrane domain of CD83 as a key regulatory element blocking activity of MARCH-1/8 and thereby stabilizing surface display of MHC-II and CD86 [31,37]. MARCH (membrane-associated RING-CH) proteins are E3-ubiquitin ligases with copious functions in the immune system [56]. MARCH-1/8 polyubiquitinate MHC-II and CD86 on residues K225 or K267, respectively, which directs these proteins to the lysosome for degradation. Interestingly, while MARCH-1 controlled MHC-II expression in splenic DCs and B cells, it was dispensable for ubiquitination in cortical thymic epithelial cells (cTEC) [57]. By contrast, cTECs express another family member of MARCH proteins, namely MARCH-8 [37]. Via inhibition of MARCH-8 in cTECs, CD83 secures MHC-II expression and antigen presentation to developing CD4^+^ T cells, which is a prerequisite for proper positive selection. CD83 deletion causes diminished expression of MHC-II, resulting in defective CD4^+^ T cell development, and is thus responsible for drastically reduced peripheral T cell numbers [37,44]. Similarly, increased activity of MARCH-1 upon CD83 deletion in B cells and DCs causes excessive degradation of MHC-II and CD86 [31,54]. Thus, it is rather puzzling that CD83-deficient APCs are sufficiently capable of mounting antigen-specific T cell responses [29,30,52,53]. One explanation could be that altered ubiquitination of MHC-II heavily perturbs DC biology: March1^−/−^ DCs, which exhibit excessive surface expression of MHC-II and CD86, secrete less IL-12 and show impaired potential to stimulate CD4^+^ T cells, probably due to ineffective peptide loading [58,59,60]. Thus, March1^−/−^ and CD83-deficient DCs exhibit the exactly opposite phenotype. Consequently, MHC-II turnover and peptide loading might be enhanced in CD83-deficient DCs, leading to improved T cell stimulatory capacity. Furthermore, engagement of CD83 with antibodies dampened p38 MAPK signaling in maturing DCs [29], and specific deletion of CD83 in DCs results in elevated surface expression of OX40L, probably compensating for reduced CD86 expression [30].

When it comes to identification of further CD83-induced signaling pathways, one major complicating circumstance is owed to a lack of consensus sequences in the cytoplasmic tail, which allow for recruitment of adaptor molecules. A very recent study demonstrated that the cytoplasmic tail of CD83 associates with TAK1 and TAB1 in human ovarian cancer cells, by which it elicits pro-tumorigenic ERK1 signaling [61]. Next to secretion of sCD83 to mediate immune evasion, this signaling pathway could be the reason for CD83 expression of several hematologic malignancies [43]. To what extent ERK1 signaling is also modulated in immune cells, remains yet to be established.

Collectively, CD83 is indispensable for proper T cell selection, influences antigen presentation by inhibiting MARCH-dependent ubiquitination of MHC-II and CD86, and can affect cellular signaling pathways. This highlights its importance for induction and modulation of immune responses.

#### 2.1.2. Signaling of sCD83—A Tale of Two Pathways

Regarding sCD83, recent advances narrowed its signaling events down to the modulation of two important pathways in immune responses: (i) interference with the TLR4/MD-2 complex and (ii) activation of a TGF-β/indoleamine-2,3-dioxygenase (IDO1) axis. Since its discovery in the sera of leukemia patients, where it correlates with reduced progression-free survival [25,62], sCD83 is considered as a potent immunomodulatory agent. The addition of sCD83 to co-cultures of DCs and T cells inhibits T cell proliferation [63], and release of sCD83 from lymphoma cells blocks antitumor T cell responses [43]. Paralleling this in vitro data, sCD83 proved its efficacy in several pre-clinical animal models of autoimmunity and transplantation by reducing or preventing disease progression, severity, or transplant rejection [45,47,51,64,65,66,67,68,69]. Despite this clear evidence for its immunomodulatory effects, sCD83-mediated signaling had been elusive for a long time, despite the fact that several immune cell types have been proposed to express potential receptors for sCD83 (reviewed in [19]). Therefore, elucidation of sCD83-mediated signaling advanced considerably with the identification of MD-2 as a binding partner on human monocytes [70]. MD-2 is an integral component of the LPS-sensing TLR4-complex [71], and its interaction with sCD83 required accessory proteins of this complex, namely CD14 and CD44v6. Further, treatment of monocytes with sCD83 interferes with the pro-inflammatory TLR-4 signaling and induces an anti-inflammatory and pro-resolving phenotype in the affected cell type.

In line with this, two recent reports have demonstrated that CD83 derived from epithelial cells of the airway or intestine suppresses T cell responses in models of asthma and food allergy [38,39], and that this effect relies on intact TLR4-/MD-2 signaling. Both studies suggest that activation of the CD83/TLR4 axis results in Treg induction by enhancing expression of Krüppel-like factor 10 (Klf10/TIEG-1), a transcription factor which can modulate TGF-β expression [72]. Since in vivo immunomodulatory effects of sCD83 also (partially) rely on TGF-β-signaling [51,64], those observations can explain how sCD83 induces TGF-β. These findings also bridge the interaction of sCD83 with TLR4 and another important aspect of its immunomodulatory and pro-resolving functions, i.e., the induction of IDO1 activity. Interestingly, topical as well as systemic application of sCD83 or TGF-β significantly prolonged graft survival in an allogeneic cornea transplantation model, and the effect of both compounds was reversed upon inhibition of IDO-1 enzymatic activity by 1-methyl tryptophan (1-MT) [64]. These observations were confirmed in several pre-clinical models for corneal and liver allograft transplantation as well as for colitis and rheumatoid arthritis [47,51,68,69]. The interplay between sCD83, TGF-β, and IDO-1 relies on a regulatory, amplifying feedback loop between TGF-β and IDO-1, which depends on the inherent signaling capacity of IDO-1. Thereby, TGF-β stabilizes IDO-1 phosphorylation, leading to non-canonical NF-κB activation, and subsequent transcription of *Ido1* and *Tgfb* mRNA (reviewed in [73,74]). The importance of this IDO-1/TGF-β loop for sCD83 signaling is highlighted by the fact that, on the one hand, IDO-1 blockade with 1-MT inhibits sCD83-mediated TGF-β induction [51,69], and on the other hand, antibody-blockade of TGF-β prevented sCD83-induced expression of IDO-1 [64,69].

Furthermore, enzymatic IDO-1 activity has also been reported in human peripheral blood mononuclear cells (PBMCs) that were stimulated with antibodies against CD3 and CD28 in the presence of sCD83. In this setting, sCD83-elicited IDO-1 signaling is part of an axis via PGE_2_ and IL-10, which impairs T cell proliferation [70,75]. Collectively, these data imply a regulatory circuit by which sCD83 modulates TLR4-signaling to elicit elevated IDO-1 activity and TGF-β secretion either separately, consecutively, or concomitantly. Eventually, these pathways culminate in increased Treg induction [47,64], less maturation of DCs [63], and promotion of the resolution phase of inflammation [51].

A different aspect of sCD83 signaling involves rearrangement of the cytoskeleton in DCs. Treatment of DCs with sCD83 leads to a drastic loss of their eponymous cellular protrusions and a rounded shape [76]. They also fail to form clusters with T cells, which recently was attributed to impaired Rab1a-mediated, actin-dependent rearrangement of co-stimulatory molecules at the immunological synapse and concomitant reduced calcium release in DCs and T cells [49,77]. This pathway provides a mechanism independent of the IDO-1/TGF-β axis, by which sCD83 disturbs inflammatory processes and promotes resolution of inflammation.

### 2.2. Role of mCD83 in the Resolution of Inflammation

As mentioned above, the deleterious effect of complete CD83-deletion on CD4^+^ cell development has impeded clear predictions of the biological function of CD83 in inflammation for a long time. Due to their lack of peripheral CD4^+^ T cells, CD83^−/−^ mice show reduced responses in a contact hypersensitivity model, which is dependent on proper T cell reaction [44,55], and their remaining T cells are hyperresponsive to stimulation [55].

Thus, employing these mice does not allow evaluation of the relevance of CD83 expression on different cell types for immune responses. To circumvent this problem, we generated mice where CD83 can be deleted by the Cre-LoxP system to enable investigations on its cell-specific biologic functions. Preliminary studies have revealed that CD83 deletion in B cells interferes with the proper formation of germinal center reaction and antibody production in response to bacterial infection [54]. Further data on conditional deletion of CD83 in DCs and Tregs have disclosed its vital role for the resolution of inflammation, which we will discuss in the following section.

#### 2.2.1. CD83 Expression by DCs—Fine-Tuning of Inflammation and Its Resolution

Two recent studies, which used conditional KO (cKO) strategies, have shed light on the regulatory function of CD83 expressed by DCs in the context of chronic and autoimmune inflammation as well as resolution of inflammation. The first study has revealed that specific deletion of CD83 expression in DCs aggravates disease symptoms in chemical- and bacterial-induced colitis models [29]. Moreover, the authors reported that overexpression of CD83 on intestinal epithelial cells mitigated DC activation and thus, severity of colitis [29]. Further, they demonstrated that CD83-deletion on DCs enhanced secretion of IL-12, a cytokine crucial for anti-bacterial immunity. The second study also reported elevated expression of IL-12 in CD83-deficient DCs, alongside with improved bacterial clearance [30]. Interestingly, the authors have further highlighted that deletion of CD83 imposes an over-activated phenotype on DCs, which is characterized by elevated secretion of IL-2, enhanced expression of co-stimulatory molecules (i.e., CD25 and OX40L), and consequently, improved antigen-specific T cell stimulatory capacity. In vivo, this over-activation culminates in augmented inflammation by subverting Treg-mediated resolution of inflammation in CD83 cKO mice in the experimental autoimmune encephalomyelitis (EAE) model. These studies not only revealed that CD83 is an important checkpoint tuning inflammatory processes, which fine-tunes DC activation during inflammation and is indispensable for its attenuation during the resolution.

#### 2.2.2. CD83 Expression by Tregs—Gate-Keeper of Tolerance

Previous studies have shown that CD83 is not only expressed by DCs but also by activated lymphocytes, including B and T cells [18]. Although the observed upregulation of CD83 was mainly attributed to B cells, several further reports have demonstrated CD83 expression on T cells after specific stimulation [78,79,80]. Interestingly, while effector CD4^+^ T cells gradually lose CD83 expression after activation, Tregs show prolonged surface expression [81,82]. Furthermore, retrovirus mediated CD83 overexpression in naïve CD4^+^ T cells converts them into a Treg-like phenotype with suppressive capacities in vivo [82]. Furthermore, ectopic expression of human CD83 in mice ameliorated autoimmune neuro-inflammation, mainly due to enhanced Treg activity and less proliferation of T effector cells [83]. Interestingly, CD83^+^ lymphoma cells can transfer mCD83 onto surrounding T cells, thereby subverting their anti-tumor activity [43]. More extensive analyses of CD83-expressing T cells revealed that CD83^+^/CD4^+^ T cells selectively express high levels of Treg associated surface markers and conferred immunosuppression in a transfer colitis model [80].

Since these data suggest a crucial role of CD83 for Treg biology and function, it is not surprising that specific deletion of CD83 in either Tregs or CD4^+^ T cells critically impairs their suppressive capacities [36,84]. Thereby, Treg-specific ablation of CD83 enhances autoimmune reactions, as demonstrated by elevated anti-nuclear antibodies, aggravated disease course in the EAE model as well as exacerbated colitis symptoms [36]. Transcriptomic analyses revealed that Tregs from cell-specific CD83 KO animals exhibit an aberrant activation status, characterized by reduced differentiation and homing markers, i.e., KLRG1 and CD103, respectively [36]. Compared to wild-type cells, CD83-deficient Tregs produce more pro-inflammatory cytokines, which can be attributed to elevated expression of components of the TLR4-signaling pathway. As sCD83 also modulates TLR4 signaling, these data highlight the importance of the sCD83/TLR4 axis in the resolution of inflammation. Since CD83-deficent Tregs and DCs exhibit an over-activated status, mCD83 is necessary to limit inflammatory processes, rendering it a critical target for pro-resolution therapies.

In summary, mCD83 deletion results in pro-inflammatory effects, impairing resolution of inflammation. Furthermore, administration of sCD83 directly leads to resolution of inflammation, indicating a pro-resolving function for both isoforms. In Figure 1 we summarize the data on known CD83-induced signaling events, which contribute to resolution of inflammation in different cell types.

## 3. Clinical Relevance of sCD83 for Therapeutic Purposes

There is an extensive body of evidence, demonstrating that sCD83 administration is a potent means to promote pro-resolution effects in preclinical disease models. Thus, in the following chapter, we will summarize the beneficial effects induced by sCD83 in these models, with a special focus on chronic inflammatory and autoimmune conditions as well as strategies to prevent organ transplant rejections.

### 3.1. sCD83 Promotes the Resolution of Chronic Inflammation

Inflammation is a tightly controlled physiological process of sequentially activated defense mechanisms. Imbalanced inflammation can lead to a misdirected anti-self-reaction that manifests as autoimmune disease, marked by chronic inflammation, destruction of healthy tissue and a loss of tissue functionality. Conventional therapy commonly relies on anti-inflammatory and immunosuppressive drugs to treat overshooting immune reactions. However, the low response rate and strong side effects of these therapies are not satisfying. Thus, new therapeutic concepts aim at modulating/manipulating cells involved in the autoimmune activation process or induce regulatory mechanisms to overcome the disease by enforcing the resolution process. Several independent groups have already proven the efficacy of sCD83 as such a “pro-resolution-compound” in preclinical models for chronic inflammatory and autoimmune diseases [39,45,46,47,48,50,51], which we will now discuss in more detail.

#### 3.1.1. Autoimmune Disorders

Preliminary investigations regarding the anti-inflammatory and/or pro-resolving function of sCD83 revealed that administration of sCD83 alleviated the disease course of EAE, which is an animal model for the early phase of human multiple sclerosis [45]. In this study, prophylactic treatment with sCD83 reduced inflammatory cell influx into the central nervous system, and importantly, sCD83 exhibited its ameliorating activity even when applied after disease onset, highlighting its resolving capacity also in a therapeutic setting. Interestingly, sCD83 is elevated in the sera of MS patients and its levels are negatively correlated with disease activity in these patients [26]. Given the importance of IDO-1 for the remission of MS and EAE (reviewed in [85]) and the striking data on the sCD83/IDO-1 axis (see above), it is conceivable that sCD83 secretion represents a counter-mechanism in MS pathology, acting via IDO-1.

Additionally, treatment with sCD83 induced the resolution of inflammation via the reduction of inflammatory cell infiltration, induction of tolerogenic DCs, and generation of regulatory NK cells in a model of autoimmune uveitis [48,49]. Mechanistically, sCD83 induces tolerogenic DCs by decreasing the synaptic expression of co-stimulatory molecules and decreases F-actin-dependent calcium signaling in DCs [77], which was also confirmed by an independent study. These changes are caused by a disruption of the cytoskeletal rearrangements at the DC-T cell contact zone, leading to altered localization of calcium micro-domains and suppressed T-cell activation, by which sCD83 promotes the resolution of inflammation in autoimmune uveitis [49]. Collectively, sCD83 treatment induces regulatory cells, which are crucial for re-establishing normal physiology after inflammation.

#### 3.1.2. Inflammatory Bowel Disease

Inflammatory bowel diseases (IBD), such as Crohn’s disease and ulcerative colitis, are severe inflammatory conditions that preferentially affect the distal small intestine or colon, with little signs of spontaneous resolution. Tissue-targeted immune reactions, as observed upon intestinal infection or in IBD, pose a vital threat to the organism and are often a result of disturbed homeostatic immune mechanisms and intestinal dysbiosis [86,87]. Tissue-resident immune cells e.g., intestinal DCs and MΦs and their mutual interactions with intestinal epithelial cells (IECs) of the gut are crucial for the maintenance of intestinal immune homeostasis. Both cell types do not only contribute to steady state physiology, but also to the resolution of inflammation, once the inflammatory stimulus is cleared. Interestingly, intestinal DCs promote the conversion of inflammatory T cells into immunosuppressive Tregs in an IDO-dependent manner, enabling induction of immune tolerance in the gut [88]. Experimental models demonstrated that IDO1 promotes gut immune homeostasis by limiting inflammatory responses and protecting the epithelium [89]. Interestingly, IDO expression in mesenteric lymph nodes was enhanced by sCD83 during DNBS induced colitis, and sCD83 administration protected from severe colitis symptoms by inhibiting inflammatory cell influx and inflammatory cytokine production [47]. Notably, blockade of IDO activity by 1-MT increased DNBS-induced mortality and clinical symptoms in sCD83-treated mice. Moreover, sCD83 promoted resolution of inflammation and tissue repair since sCD83-treated mice showed an intact colonic architecture. Growing evidence has indicated that dysfunctional TLR-4-signaling plays a pivotal part in the pathogenesis of IBD [90]. Since sCD83 modulates the TLR-4 axis to promote pro-resolving mechanisms, IBD patients with elevated TLR-4 expression levels on intestinal cells might especially profit from a future sCD83 treatment. Thus, sCD83 could be used in targeted pro-resolution therapy of IBD patients.

#### 3.1.3. Rheumatoid Arthritis

Rheumatoid arthritis (RA) is a chronic inflammatory disorder affecting the joints, which causes painful swelling and eventually results in bone erosion and joint deformation. Severe courses also show systemic manifestations, which affect the skin, eyes, lungs, heart, and blood vessels [91]. Standard treatment of this autoimmune disease focuses on controlling the autoimmune reaction with immunosuppressant medications like corticosteroids. Furthermore, current therapeutic approaches are directed against pro-inflammatory effector cytokines, such as TNF-α, IL-6, IL-1, IL17, IL-12/IL-23, IL-5, or IL-4/13 using monoclonal antibodies or drugs against pro-inflammatory cytokine signaling pathways, like Janus kinases [91]. However, such strategies, which require life-long application, bear the risk of blunting physiological immune responses and are associated with more frequent infections. Thus, RA patients would greatly benefit from pro-resolution therapies that establish long-lasting tolerance.

In this regard, treatment of mice with sCD83 effectively induces a significant reduction of disease-associated symptoms in both antigen-induced and chronic arthritis models [51]. Mechanistically, sCD83 triggers the resolution of inflammatory arthritis via downregulation of inflammatory cytokines (IFN-ɣ, TNF-α, IL-17a and IL-6) within the joints. Joint swelling as well as bone erosion was significantly reduced in sCD83 treated arthritic mice, which is attributed to reduced expression of receptor activator of nuclear factor-κB ligand (RANKL), a key regulator of osteoclast differentiation. Additional in vitro data revealed that sCD83 interferes with osteoclast formation/differentiation, resulting in attenuated bone and cartilage destruction. As in IBD, sCD83 induced the resolution of inflammation via the IDO-1 pathway, since inhibition of IDO-1 enzymatic activity by 1-MT completely abrogated the sCD83 mediated effects. IDO-1 expression in myeloid cells has been shown to induce the generation of regulatory DCs and inhibit differentiation of osteoclasts [92]. Interestingly, a recent study on RA patients revealed that expression levels of *CD83* and *IDO1* were elevated in the blood of those patients who responded to therapy with abatacept (a fusion protein of CTLA-4 and IgG1, which affects DC biology and T cell stimulation) [93]. Together with the observation that the synovial fluid of RA patients contains elevated levels of sCD83 [27], these data clearly demonstrate the translational importance of the CD83/IDO1-axis.

Next to IDO-1, sCD83 treatment induces the expression of TGF-β, and blockade of TGF-β partially reversed the anti-arthritic effects of sCD83. As mentioned above, IDO1 and TGF-β form a mutually enforcing circuit, which is the mechanistic basis for the observed sCD83-mediated pro-resolving effects. Since administration of sCD83 also increases the frequency of regulatory T cells, it promotes long-term tolerance, which prevents disease relapses even in the absence of continuous treatment. Collectively, the sCD83 protein represents a promising therapeutic treatment option to resolve autoimmune mediated disorders, since it directly counters the impaired resolution process and promotes resolution of inflammation.

### 3.2. sCD83 Prevents Graft Rejection by Induction of Tolerogenic Mechanisms

In contrast to autoimmune diseases, in which the immune system reacts to self-antigens, unwanted inflammatory responses of innate and adaptive immune cells occur in the field of transplantation. Rejections of allografts occur due to fatal immune reactions of the recipient to the donor tissue. These inflammatory responses rely on differences of highly polymorphic MHC molecules between recipient and donor, resulting in tissue damage and finally rejection of the transplanted tissue. Current therapeutic approaches in patients receiving organ transplants often rely on a non-specific immunosuppressive medication, such as glucocorticoids, cytostatics, calcineurin inhibitors, or mTOR inhibitors with all known associated negative side effects [94]. Thus, patients after organ transplantation often suffer from drug-associated toxicity, reduced resistance to infections and development of malignancies. Consequently, new therapeutic agents, which establish or induce immune tolerance, promote tissue repair, contribute to resolution of inflammation are urgently needed. For this, researchers pursue amongst others the following strategies: (i) induction or transfer of Tregs, which are able to induce immune tolerance, and (ii) modulation of APC populations including DCs as well as Mφ towards regulatory cells, which can promote Treg differentiation and thus, induce immune tolerance. In this respect, regulatory Mφ, DCs, as well as Tregs, which are able to resolve inflammatory responses have been used in clinical trials as a cellular therapy in combination with immunosuppressive drugs in kidney transplantation [95].

Another strategy is represented by the use of immune checkpoints to specifically modulate these regulatory immune cells. In recent years, using different preclinical rodent transplantation models, several groups have demonstrated that the sCD83 molecule exactly meets these criteria to induce immune tolerance. Systemic administration of sCD83 significantly prevented the rejection of skin, heart, kidney, and liver transplants [19,65,66,67,69,96]. Resolution of inflammation and induction of graft tolerance by systemic sCD83 administration was achieved via the TGF-β and IDO1 pathway leading to the induction of Tregs, e.g., in kidney and liver transplantation models [66,69]. In addition to systemic treatment, topical application of sCD83 in the form of eye-drops remarkably prolongs allograft survival in a fully mismatched high-risk corneal transplantation model [64]. Administration of TGF-β blocking antibodies or the IDO1-inhibitor 1-MT completely abolish sCD83 induced allograft survival, further elucidating the mechanistic background of sCD83-mediated effects on this regulatory circuit [64]. Recently, a new therapeutic concept of transplant-mediated tolerance induction using the sCD83 molecule has been established, namely the pre-incubation of donor corneal tissue with sCD83. Surprisingly, this pre-incubation of grafts is already sufficient to induce resolving mechanisms within graft recipients and significantly prolongs allograft survival [68]. Tolerance induction via the pre-incubation of donor tissue with sCD83 depends on increased frequencies of CD4^+^Foxp3^+^ Tregs in eye draining lymph nodes as well as upregulation of immunoregulatory mediators within the graft, such as IDO-1, IL-27, and IL-10. Moreover, the preincubation of donor grafts with sCD83 sculpts a tolerogenic DC phenotype in vivo, characterized by a shift from immunogenic CD80^+^ DCs to CD200R^+^ regulatory DCs [68]. These immunoregulatory effects of sCD83 on DC biology have been proven in earlier studies in the context of kidney, heart, liver, and cornea transplantation [64,65,66,69]. Here, sCD83 induces IDO1^+^ DCs, which are capable to promote Tregs, which contribute to resolution of inflammatory responses. This again highlights the crucial importance of IDO1 in sCD83-mediated pro-resolving effects.

Interestingly, the recent study unveiled that sCD83 also affected Mφ phenotypes in corneal transplantation [68]. Mφ are a heterogeneous cell population and can specifically acquire different phenotypes and functions. Thus, modulation of Mφ represents an interesting approach to enhance graft acceptance, since pro-inflammatory CAMs (TNF-ɑ^+^, IL-6^+^, CD86^high^) are involved in transplant rejection whereas regulatory AAM (MSR-1^+^, CD86^low^, CCL-22^+^) are cells with strong anti-inflammatory properties involved in immune regulation, tissue remodeling, resolution of inflammation and transplant acceptance. Furthermore, AAM-like regulatory Mφ are able to expand CD4^+^Foxp3^+^ T cells and control effector T cell fate [97], and Mφ that have been co-cultured with Tregs display typical features of AAM [98]. Notably, upon pre-incubation of donor tissue with sCD83, this molecule has been found to be co-localized with F4/80^+^ Mφ in the donor tissue as well as in the eye-draining lymph nodes of the recipient. In addition, transplantation of sCD83 pre-treated grafts resulted in a decreased frequency of CAMs and an increased frequency of AAMs in eye-draining lymph nodes [68].

Further in vitro studies revealed that the presence of sCD83 during DC and Mφ differentiation directed these APC populations toward a tolerogenic phenotype [68]. Indeed, administration of sCD83 during Mφ differentiation modulated them towards a pro-resolving AAM phenotype and function, resulting in the secretion of CCL17 as well as CCL22, both chemokines known to be important for Tregs recruitment [99]. This sCD83 induced resolving phenotype was further characterized by the downregulation of the costimulatory molecule CD86 on Mφ, whilst pro-resolving markers such as Msr-1 were strikingly upregulated, (see Figure 2). DCs, which were differentiated in the presence of sCD83, presented similar phenotypic changes. On a functional level, these Mφ and DCs inhibited T cell proliferation and promoted the induction of Foxp3^+^CD4^+^ T cells [68]. In order to investigate the immune regulatory properties of in vitro sCD83-treated Mφ to induce tolerance in vivo, adoptive transfer experiments were performed in a fully MHC mismatched high-risk corneal transplantation model. Strikingly, administration of sCD83-treated regulatory iDCs or Mφ improves the survival of cornea grafts [68]. The sCD83 induced pro-resolving changes on Mφ are summarized in Figure 2.

## 4. sCD83: Conclusions, New Insights and Future Direction

In recent years, the CD83 molecule has been identified as an important immunological checkpoint, which contributes to the resolution of inflammation. The mCD83 isoform controls inflammatory responses via the induction of regulatory mechanisms in DCs as well as Tregs, while the sCD83 isoform induces resolution of inflammation in autoimmunity and promotes tolerance induction after transplantation. Importantly, a recent study paved the way for new therapeutic options in the field of transplantation: (i) pre-incubation of donor grafts with sCD83 is sufficient to induce tolerogenic mechanisms after transplantation, and (ii) adoptive transfer of sCD83-treated APCs improves graft survival. Since both regulatory DCs and Mφ have already been used in clinical trials after transplantation, the prospect of using sCD83-induced regulatory cells in transplantation is very exciting. Furthermore, this strategy could be combined with direct administration of the pro-resolving sCD83 molecule to modulate pro-inflammatory immune responses and to further promote resolution of inflammation. Similarly, sCD83-treated NK cells conferred immune modulation and disease amelioration in the EAU setting [48], highlighting the potency of sCD83 to boost cell-based therapy approaches. Regarding the role of mCD83 as pro-resolving mediator, adoptive transfer of antigen-specific T cells ectopically over-expressing mCD83 represents another very interesting approach. Collectively, further exploration of the potential of CD83 as a “pro-resolution” therapy may provide modern healthcare with an interesting tool to combat chronic and autoimmune diseases as well as to improve transplant acceptance.

## Figures and Tables

**Figure 1 ijms-23-00732-f001:**
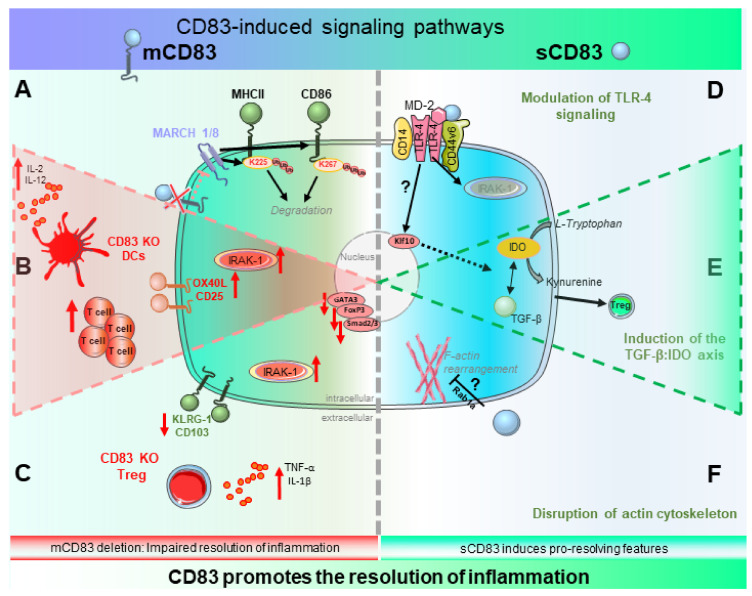
Schematic model depicting the signaling pathways of mCD83 (left side, (**A**–**C**)) as well as sCD83 (right side, (**D**–**F**)), which contribute to resolution of inflammation. (**A**) The transmembrane domain of mCD83 stabilizes MHCII (and CD86) by antagonizing MARCH dependent ubiquitination in TECs and immune cells. (**B**) Deletion of mCD83 in DCs results in an over-activated DC phenotype. DCs show enhanced expression of costimulatory OX40L as well as CD25 and an increased secretion of the pro-inflammatory cytokine IL-12 as well as IL-2 upon stimulation with TLR-ligands. This correlates with increased IRAK-1 levels and concordantly, CD83-deficient DCs stimulate T cells more potently (up-facing red arrows) (**C**) CD83 deletion in Tregs results in a pro-inflammatory phenotype, characterized by decreased expression levels of Gata3, Foxp3, Smad2/3 as well as KLRG1 and CD103 (down-facing red arrows). In addition, CD83-deficient Tregs show increased expression levels of TLR-2 and TLR-4, IRAK-1 as well as secretion of pro-inflammatory cytokines e.g., TNF-α and IL-1β. (**D**) Binding of sCD83 to the MD-2/TLR-4 complex leads to IRAK-1 degradation which results in anti-inflammatory responses. In T cells, this leads to induction of Klf10. (**E**) sCD83 induces the TGF-β/IDO axis, whereby IDO leads to enhanced conversion of tryptophan into kynurenine. Mechanistically, tryptophan deprivation leads to inhibition of T effector cell proliferation and kynurenines induce expansion of Tregs via the AhR pathway. Upregulation of TGF-β by sCD83 induces the IDO1-signaling activity, which in turn leads to prolonged induction of Tregs. (**F**) Via Rab1a, sCD83 interferes with rearrangement of actin-filaments, modulating the density of co-stimulatory molecules at the immunological synapse. Figure was created using images from https://smart.servier.com/ (accessed on 29 November 2021) with adaptations.

**Figure 2 ijms-23-00732-f002:**
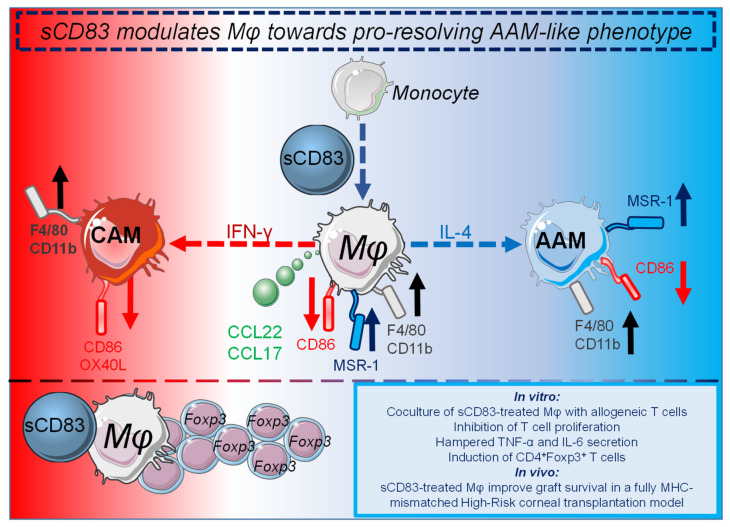
sCD83 modulates Mφ towards an AAM-like phenotype with pro-resolving functions [68]. Administration of sCD83 during Mφ differentiation results in the secretion of AAM-associated chemokines CCL22 and CCL17, which are important for the recruitment of Tregs. In addition, F4/80, CD11b (black arrows) and the pro-resolving Msr-1 molecule were upregulated (dark blue arrows), while the costimulatory molecule CD86 was downregulated (red arrow). Coculture of sCD83-treated Mφ with allogeneic T cells results in inhibition of T cell proliferation, induction of Tregs as well as reduced IL-6 and TNF-α secretion. Administration of sCD83 during Mφ differentiation and subsequent skewing either towards CAMs or AAMs results in a prominent downregulation of CD86, OX40L (red arrow) on CAMs and in an upregulation of MSR-1 as well as downmodulation of CD86 on AAMs. Subsequently, both sCD83treated CAMs and AAMs enhance CD4^+^Foxp3^+^ T cell frequencies in MLR cocultures. Adoptive transfer of sCD83-treated Mφ results in the induction of tolerance in a fully MHC mismatched high-risk corneal transplantation model. Thus, in conclusion, sCD83 modulates Mφ towards an AAM phenotype, which promotes the resolution of inflammation. Figure was created using images from https://smart.servier.com/ (accessed on 29 November 2021) with adaptations.

## Data Availability

Not applicable.

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
