# Peer review of "Tilting the Balance: Therapeutic Prospects of CD83 as a Checkpoint Molecule Controlling Resolution of Inflammation"

_ijms, 2022, doi:10.3390/ijms23020732_

Round 1
Reviewer 1 Report
In my opinion the paper may be published in the current form - it is a complete and very valuable and comprehensive paper on a novel candidate to a checkpoint molecule. The paper is a deep review of the literature with a nice interpreted figures attached. References are critical and well chosen.
Author Response
In my opinion the paper may be published in the current form - it is a complete and very valuable and comprehensive paper on a novel candidate to a checkpoint molecule. The paper is a deep review of the literature with a nice interpreted figures attached. References are critical and well chosen.
RE: We highly appreciate the reviewer’s positive feedback to our work. We are very pleased about the comments to the created figures describing the recent literature in CD83 research and that references are well chosen. Thank you.
Reviewer 2 Report
This review discusses the biology and applications of inhibitory molecule CD83. Given the outstanding clinical needs for autoimmunity and transplantation, the topic is of interest to the field.
The manuscript is overall well-written and clear. The rationale of the ideas outlined is both logical and sound.
A general question on section 3, therapy: does the extensive glycosylation render the clinical-grade production of sCD83 more challenging? It would be interesting to know.
A few minor points:
Pg1:
There are also likely to be resident macrophages taking part in initial responses; it could be useful to insert them in the narrative.
Pg2
The DC that end up inducing tolerance are unlikely to be the same as those that initiated the response. Thus the phrase “These cells not only are responsible for mounting a protective T cell response against invading pathogens but also promote and maintain immunological tolerance by induction of regulatory T cells “ is not very precise. The tolerogenic DC could be local, but they could also be intestinal DC inducing iTreg, or thymic APC inducing nTreg. I understand that the authors in this segment wish to place DC at the forefront (and this is perfectly acceptable), but their statements must be valid even for an audience that may approach the text from a different perspective, so there must be no space for misinterpretations.
Pg 4
The meaning of the phrase “Thus, it is rather puzzling that CD83-deficient APCs are sufficiently capable to mount antigen-spe- cific T cell responses [29, 30, 52, 53].
One explanation could be that March1-/- DCs – despite excessive surface expression of MHC-II and CD86 – are less potent to stimulate CD4+ T cells, probably due to ineffective peptide loading” is somehow confusing. Please rephrase, with more clarity and detail.
Author Response
This review discusses the biology and applications of inhibitory molecule CD83. Given the outstanding clinical needs for autoimmunity and transplantation, the topic is of interest to the field. The manuscript is overall well-written and clear. The rationale of the ideas outlined is both logical and sound.
RE: We highly appreciate the positive feedback on our manuscript and want to thank you very much for the valuable comments and very helpful suggestions, which will improve our article.
A general question on section 3, therapy: does the extensive glycosylation render the clinical-grade production of sCD83 more challenging? It would be interesting to know.
RE: We thank you for the question regarding the glycosylation of the sCD83 protein. Based on previous studies, glycosylation is not essential for the pro-resolving sCD83-mediated biological activity. In studies on the inflammation-resolving effects of sCD83 in EAE and cornea transplantation, these groups used recombinant sCD83 produced in Escherichia coli, which lack protein glycosylation. This shows that the glycosylation stage of sCD83 is not important for pro-resolving properties. Furthermore, the use of sCD83-produced proteins in E.coli significantly inhibited DC-mediated T cell stimulation (Lechmann et. al., 2002), and similar results were obtained with sCD83, which was recombinantly produced in human HEK293 cells. Since both E.coli and HEK293 resulted in rather low yields of sCD83 protein, we used a different expression system (i.e. Pichia pastoris, as described by Guo et. al., 2014, doi: 10.1371/journal.pone.0089264) in our recent studies (Royzman et. al., 2019, Peckert-Maier et. al., 2021). Since yeast cells possess a different glycosylation machinery than mammalian cells, sCD83 proteins from Pichia pastoris might have a different glycosylation pattern. Nonetheless, sCD83 expressed in Pichia pastoris has equally potent pro-resolving capacities. In conclusion, glycosylation of sCD83 is not essential to induce resolution of inflammation, which facilitates its production for therapeutic purposes
A few minor points: Pg1: There are also likely to be resident macrophages taking part in initial responses; it could be useful to insert them in the narrative.
RE: We thank you for this comment. Of course, resident macrophages are taking part within initial immune response. We changed the text accordingly on page 1 as shown below:
Revised Page 1: Tissue resident cells, such as macrophages (Mφ), sense an inflammatory stimulus via pattern-recognition receptors, such as toll-like receptors (TLRs), which causes activation of pro-inflammatory signaling cascades, such as the nuclear factor kappa B (NF-κB) pathway [1].
Pg2: The DC that end up inducing tolerance are unlikely to be the same as those that initiated the response. Thus the phrase “These cells not only are responsible for mounting a protective T cell response against invading pathogens but also promote and maintain immunological tolerance by induction of regulatory T cells “ is not very precise. The tolerogenic DC could be local, but they could also be intestinal DC inducing iTreg, or thymic APC inducing nTreg. I understand that the authors in this segment wish to place DC at the forefront (and this is perfectly acceptable), but their statements must be valid even for an audience that may approach the text from a different perspective, so there must be no space for misinterpretations.
RE: We thank you for this comment and of course we would like to avoid misinterpretations. Therefore, we changed the text to clarify that specific subsets of DCs can maintain or induce tolerance by induction of Tregs.
Revised Page 2: DCs function as control center at the interface between innate and adaptive immunity and therefore, fine-tune initiation and eventual confinement of inflammatory processes. Furthermore, specific subsets of these cells promote and maintain immunological tolerance by induction of regulatory T cells (Treg), and thus can restrain inflammation [4].
Pg 4: The meaning of the phrase “Thus, it is rather puzzling that CD83-deficient APCs are sufficiently capable to mount antigen-specific T cell responses [29, 30, 52, 53]. One explanation could be that March1-/- DCs – despite excessive surface expression of MHC-II and CD86 – are less potent to stimulate CD4+ T cells, probably due to ineffective peptide loading” is somehow confusing. Please rephrase, with more clarity and detail.
RE: We thank the reviewer for the comment and apologize for the inaccuracy and confusing statements. According to the reviewer’s suggestion, we rephrased the text to provide more clarity and detail.
Revised Page 4: One explanation could be that altered ubiquitination of MHC-II heavily perturbs DC biology: March1-/- DCs, which exhibit excessive surface expression of MHC-II and CD86, secrete less IL-12 and show impaired potential to stimulate CD4+ T cells, probably due to ineffective peptide loading [58-60]. Thus, March1-/- and CD83-deficient DCs exhibit the exactly opposite phenotype.
Reviewer 3 Report
This is a detailed and convincing review. It is well structured and takes into account current publications. The paragraphs cover the different aspects and possible applications and Clinical relevance of sCD83 for Therapeutic Purposes. However it would be important in 3.1.3. Rheumatoid arthritis that the authors reported and commented on recent bibliography:
Yokoyama-Kokuryo W, et al. Identification of
molecules associated with response to abatacept in patients with rheumatoid arthritis. Arthritis Res Ther. 2020 Mar 12; 22 (1): 46.
Bing N, et al Contribution of a European-Prevalent Variant near CD83 and an East Asian-Prevalent Variant near IL17RB to Herpes Zoster Risk in Tofacitinib Treatment: Results of Genome-Wide
Association Study Meta-Analyzes. Arthritis Rheumatol. 2021 Jul; 73 (7): 1155-1166
Author Response
This is a detailed and convincing review. It is well structured and takes into account current publications. The paragraphs cover the different aspects and possible applications and Clinical relevance of sCD83 for Therapeutic Purposes. However it would be important in 3.1.3. Rheumatoid arthritis that the authors reported and commented on recent bibliography:
Yokoyama-Kokuryo W, et al. Identification of molecules associated with response to abatacept in patients with rheumatoid arthritis. Arthritis Res Ther. 2020 Mar 12; 22 (1): 46.
Bing N, et al Contribution of a European-Prevalent Variant near CD83 and an East Asian-Prevalent Variant near IL17RB to Herpes Zoster Risk in Tofacitinib Treatment: Results of Genome-Wide
Association Study Meta-Analyzes. Arthritis Rheumatol. 2021 Jul; 73 (7): 1155-1166
RE: We thank the reviewer for these very positive comments and helpful suggestions on additional references. We gladly incorporated the study of Yokoyama-Kokuryo W, et al. in the respective section. This supports the major conclusions of our review article.
The second recommended reference demonstrates association of a genetic variant near the CD83 locus with reactivation of Herpes zoster infection under tofacitinib treatment. However, section 3.1.3 deals with the impact of soluble CD83 on inflammatory diseases. Thus, in our opinion, this reference lies not within the scope of our manuscript, especially since it does not show direct involvement of CD83 in the viral reactivation process.